# Ablation Path Saliency

## Abstract

We consider the saliency problem for black-box classification. In *image* classification, this means highlighting the part of the image that is most relevant for the current decision. We cast the saliency problem as finding an optimal ablation path between two images. An ablation path consists of a sequence of ever smaller masks, joining the current image to a reference image in another decision region. The optimal path will stay as long as possible in the current decision region. This approach extends the *ablation tests* in Sturmfels et al. (2020). The gradient of the corresponding objective function is closely related to the *integrated gradient method* Sundararajan et al. (2017). In the saturated case (when the classifier outputs a binary value) our method would reduce to the *meaningful perturbation approach* Fong & Vedaldi (2017), since crossing the decision boundary as late as possible would then be equivalent to finding the smallest possible mask lying on the decision boundary. Our interpretation provides geometric understanding of existing saliency methods, and suggests a novel approach based on ablation path optimisation.

## 1 Introduction

The basic idea of *saliency* or *attribution* is to provide something from which a human can judge how a classifier arrived at its decision of the prediction it gives for a certain input. It is difficult to give a more mathematical definition, but various properties that such a method should fulfill have been proposed.

Sundararajan et al. (2017) give axioms, of which *sensitivity* comes closest to the notion of saliency. Essentially, the features on which the output is most sensitive should be given a higher saliency value. The authors give further axioms to narrow it down – implementation invariance, completeness, linearity and symmetry-preservation – and obtain a corresponding method: the *integrated gradient* method. Note that we have another way to arrive at a similar method, see §4.1

Fong & Vedaldi (2017) is closer to our work: the authors directly compute the saliency of a given pixel by deleting, or altering that pixel, to see how this affects the output of the classifier.

Our method is to define a proper maximisation problem as follows. First, we define *ablation paths* as time dependent smooth masks $\varphi \colon [0, 1] \to \mathcal{C}(\Omega, \mathbb{R})$, going a full mask to an empty mask, such that at each pixel the mask value decreases over time (see Figure 1). We also impose constant area speed:

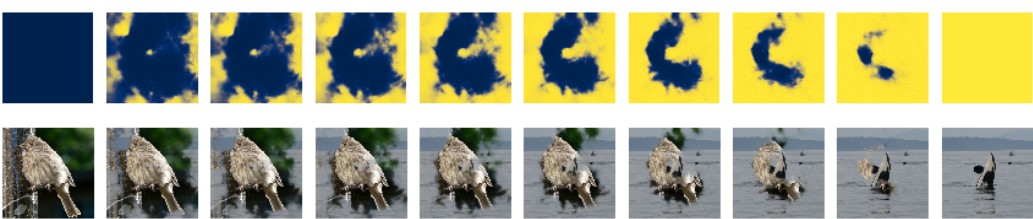

Figure 1: Example of how an ablation path (sequence of masks, top) gives rise to a transition between a current target (image of a house finch) and a baseline (orca whale).

the area covered by the mask should increase linearly over time (see §3). Let $F$ be the classifier, that outputs a probability between zero and one. We choose a current image of interest $x_0$ and a *baseline image* $x_1$. The objective function is then $P(\varphi) = \int_0^1 F(x_0 + \varphi(t)(x_1 - x_0)) \, dt$ (see §4). Assuming that $F(x_0) \simeq 1$ and $F(x_1) \simeq 0$, maximising the objective function means that we try to find an ablation path that stays as long as possible in the decision region of $x_0$. Intuitively, we try to replace as many pixels of $x_0$ by pixels of $x_1$ while staying in the same class as $x_0$.

The main contribution of this paper is to formulate the saliency problem as an optimisation problem on paths. Doing so connects previous notions of saliency, notably integrated gradients Sundararajan et al. (2017) (see §4.1), the ablation tests Sturmfels et al. (2020) (see §4.2), and meaningful perturbations Fong & Vedaldi (2017); Fong et al. (2019) (see §4.3). Our formulation is also resolution invariant (it does not assume that images are defined on pixels); this allows to make a clear difference between images and their duals, for instance, which gives guidance as to where regularisation is needed.

## 2 Related Work

Simonyan et al. (2013) defines a saliency map as the gradient of the network output at the given image. This would appear to be a sensible definition, but the resulting saliency is very noisy because the network output is roughly constant around any particular image. Selvaraju et al. (2016) improves the situation by computing the gradient after each layer instead. This is, however, not a black-box method such as the one we propose. Koh & Liang (2017) computes an *influence function*, that is, a function that measures how the parameters would be changed by a change in the training data. Although it is a black-box method, it is not a saliency method per se. They use the gradient of the network output to find the pixel most likely to have a high saliency. The pixel that have most effect are given a higher saliency. By contrast, Petsiuk et al. (2018) proposes to directly evaluate the saliency by finding out which pixels most affect the output, similarly to Fong & Vedaldi (2017), but without using any gradients.

There are a number of meta-studies of saliency methods. Adebayo et al. (2018) lists essential properties, for instance the requirement that the results should depend on the training data in a sense that perturbing *model parameters* should change the saliency. Kindermans et al. (2017) proposes a number of property that saliency methods should satisfy. Ancona et al. (2017) compares several saliency methods and proposes a method to evaluate them (the sensitiviy-$n$ property).

## 3 Ablation Paths

### 3.1 Images and Masks

We consider a compact domain $\Omega$. Note that $\Omega$ may be discrete or continuous: in fact, we assume that $\Omega$ is endowed with a measure which could be the discrete measure (if $\Omega$ is a set of pixels) or the Lebesgue measure (if $\Omega$ is a domain in $\mathbb{R}^2$, for instance). In the sequel, $\int_\Omega$ denotes integration with respect to that measure. Without loss of generality, we assume the mass of that measure to be one, i.e., $\int_\Omega 1 = 1$.

We consider a module $M$ of functions on $\Omega$ with values in a vector space $V$ (the dimensions of $V$ represent the *channels*, and elements of $M$ represent *images*). This module is equipped with a commutative ring $R$ which represents *masks*. Concretely, in most of the paper we choose

$$M := \mathcal{C}(\Omega, V) \qquad R := \mathcal{C}(\Omega, \mathbb{R}).$$

The module structure simply means that masks can multiply images, i.e., that the operation $\theta x$ gives a new image in $M$ when $\theta \in R$ and $x \in M$, and that this operation is bilinear.

### 3.2 Ablation Paths

**Definition 3.1.** *We define the set $\mathcal{A}$ of* ablation paths *as the set of functions $\varphi \colon [0, 1] \to R$ fulfilling the following properties:*

**Boundary conditions** $\varphi(0) = 0$ *and* $\varphi(1) = 1$

**Monotonicity** $t_1 \leq t_2 \implies \varphi(t_1) \leq \varphi(t_2)$     $t_1, t_2 \in [0, 1]$

**Constant speed** $\int_\Omega \varphi(t) = t$     $t \in [0, 1]$.

*We will call* monotone paths *the paths that verify the first two conditions but not the third.*

Note that the set $\mathcal{A}$ of ablation paths is a *convex subset* of $\mathcal{L}^\infty([0, 1], R)$.

Some comments on each of those requirements are in order. (i) 0 and 1 denote here the constant functions zero and one (which corresponds to the zero and one of the algebra $R$) (ii) $\varphi(t_1) \leq \varphi(t_2)$ should be interpreted as usual as $\varphi(t_2) - \varphi(t_1)$ being in the cone of nonnegative elements[1]. (iii) If $t \mapsto \int_\Omega \varphi(t)$ is differentiable, this requirement can be rewritten as $\frac{d}{dt} \int_\Omega \varphi(t) = 1$, so it can be regarded as a *constant speed* requirement. This requirement is more a normalisation than a requirement, as is further detailed in Remark 3.3.

There is a canonical, simplest, ablation path between $x_0$ and $x_1$:

$$\ell(t) := t. \tag{1}$$

The mask is thus constant in space at each time $t$. The reader should check that all the requirements for an ablation path are fulfilled.

Note that an ablation path without the constant-speed property can always be transformed into one that does fulfil it. This is clear if the function $t \mapsto \int_\Omega \varphi(t)$ is strictly increasing, as this is then just a time reparameterisation, but this is in fact always possible, in a canonical sense. The proof is in Appendix A.

**Lemma 3.2.** *To any monotone path there corresponds a canonical ablation path.*

Since $R$ is itself a function space, an ablation path $\varphi$ is in fact a function of two arguments. In the sequel, we will abuse the notations and write $\varphi$ as a function of one or two arguments depending on the context: $\varphi(t) \equiv \varphi(t, \cdot)$. For instance, in the definition Definition 3.1 above, $\int_\Omega \varphi(t) \equiv \int_\Omega \varphi(t, \cdot) \equiv \int_\Omega \varphi(t, \mathbf{r}) \, d\mathbf{r}$.

**Remark 3.3.** *If the ablation path $\varphi$ is differentiable in time, the requirements in Definition 3.1 admit a remarkable reformulation. Define $\psi(t) := \frac{d}{dt}\varphi(t)$. All the requirements in Definition 3.1 are equivalent to the following requirements for a function $\psi \colon [0, 1] \times \Omega \to \mathbb{R}$:*

$$\psi(t, \mathbf{r}) \geq 0, \quad \int_\Omega \psi(t, \mathbf{r}) \, d\mathbf{r} = 1, \quad \int_{[0,1]} \psi(t, \mathbf{r}) \, dt = 1 \quad t \in [0, 1], \mathbf{r} \in \Omega$$

*The corresponding ablation path $\varphi$ is then recovered by $\varphi(t) := \int_0^t \psi(s) \, ds$. What this means is that differentiable ablation paths can be parameterised as densities of doubly stochastic Markov transition kernels on $[0, 1] \times \Omega$.*

## 3.3 Regularity of Ablation Paths

**Lemma 3.4.** *If $\varphi$ is an ablation path, then*

$$\|\varphi(t_1) - \varphi(t_0)\|_{\mathcal{L}^1} = |t_1 - t_0|.$$

*In particular, $t \mapsto \varphi(t, \cdot)$ is continuous as a function $[0, 1] \to \mathcal{L}^1(\Omega)$.*

*Proof.* Choose $t_0, t_1$ in $[0, 1]$. Without loss of generality, assume $t_1 \geq t_0$. Then, $\int_\Omega |\varphi(t_1) - \varphi(t_0)| = \int_\Omega (\varphi(t_1) - \varphi(t_0)) = t_1 - t_0$, from which we conclude that $\varphi(t_1) - \varphi(t_0)$ is in $\mathcal{L}^1$ and fulfils the equation above. $\square$

## 4 Score of an Ablation Path

We now fix two points $x_0$ (the *current image*) and $x_1$ (the *baseline image*) in the space of images $M$. We propose the following measure of the *score* of an ablation path (see Definition 3.1) with respect

---

[1]Here we can define the cone of nonnegative functions by $\{ f \in \mathcal{C}(\Omega, \mathbb{R}) \mid f \geq 0 \}$. In a general star algebra, this cone would be defined as $\{ x \in R \mid \exists y \in R \quad x = y^*y \}$.

to these two images. Given a *mask* $\theta \in R$, we define the *interpolated image* $[x_0, x_1]_\theta \in M$ as

$$[x_0, x_1]_\theta := (1 - \theta)x_0 + \theta x_1.$$

We now define the *score function* $P \colon \mathcal{A} \to \mathbb{R}$ from ablation paths to $\mathbb{R}$ by the integral

$$P(\varphi) := \int_0^1 F([x_0, x_1]_{\varphi(t)}) \mathrm{d}t. \tag{2}$$

Note that, as $F$ is bounded between zero and one, so is $P(\varphi)$ for any ablation path $\varphi$. The main idea is that if $F(x_0) \simeq 1$ and $F(x_1) \simeq 0$, the higher this value of $P$ is, the better the path is to describe the salient parts of the image.

Note that the function $P$ is defined regardless of the constraints placed on ablation paths, i.e., the score function $P$ is defined on the vector space of functions $\varphi \colon [0, 1] \to R$. It is straightforward to compute its differential $\mathrm{d}P$ on that space:

$$\langle \mathrm{d}P, \delta\varphi \rangle = \int_0^1 \langle \underbrace{\mathrm{d}F_{[x_0, x_1]_{\varphi(t)}}}_{\in M^*}, \underbrace{(x_1 - x_0)}_{\in M} \underbrace{\delta\varphi(t)}_{\in R} \rangle \, \mathrm{d}t.$$

So if we define the product of $D \in M^*$ and $x \in M$ producing an element in $R^*$ by $\langle xD, \varphi \rangle := \langle D, x\varphi \rangle$ as is customary, we can rewrite this differential as

$$\langle \mathrm{d}P, \delta\varphi \rangle = \int_0^1 \langle (x_1 - x_0)\mathrm{d}F_{[x_0, x_1]_{\varphi(t)}}, \delta\varphi(t) \rangle \, \mathrm{d}t.$$

Note that we know that any ablation path is bounded, so $\varphi \in \mathcal{L}^\infty([0, 1], R)$, so the differential of $P$ at $\varphi$ can be identified with the function $\mathrm{d}P_\varphi = \left[ t \mapsto (x_1 - x_0)\mathrm{d}F_{[x_0, x_1]_{\varphi(t)}} \right]$ in $\mathcal{L}^1([0, 1], R^*)$.

### 4.1 Relation with the Integrated Gradient Method

When this differential is computed on the interpolation path $\ell$ (1) and then *averaged*, then this is exactly the integrated average gradient Sundararajan et al. (2017). More precisely, the integrated gradient is exactly $\int_0^1 \mathrm{d}P_{\ell(t)} \mathrm{d}t$. Note that this is in fact an integrated *differential*, since we obtain an element in the dual space $M^*$, and this differential should be appropriately smoothed along the lines of § 5.1.

### 4.2 Relation to Pixel Ablation

Given a saliency function $\sigma \in R$ we can define a path by $\tilde\varphi(t) := \mathbf{1}_{\sigma \leq \log(t/(1-t))}$ when $t \in (0, 1)$ and define $\tilde\varphi(0) := 0$, $\tilde\varphi(1) := 1$. This path is a monotone path, except in the module of images $M = \mathcal{L}^2(\Omega, V)$, equipped with the ring of masks $R = \mathcal{L}^\infty(\Omega)$. To be an ablation path, it still needs to be transformed into a constant speed path, which is always possible as explained in Appendix A.

Note that this is a generalisation of the ablation method in Sturmfels et al. (2020). In that case, the set $\Omega$ would be a discrete set of pixels. Note that in the ranking, pixels with the same saliency would be ranked in an arbitrary way and added to the mask in that arbitrary order. In the method above, we add them all at once, but the time reparameterisation keeps that function constant longer for however many pixels were ranked the same. As long as the ranking is strict (no two pixels have the same saliency), the two methods are the same.

### 4.3 Relation to Meaningful Perturbations

In the saturated case, that is, if $F$ only takes values zero and one (or in the limit where it does), our method basically reduces to finding the interpolation with the largest mask on the boundary, in essence the approach of Fong & Vedaldi (2017). Indeed, suppose that the ablation path $\varphi$ crosses the boundary at time $t^*$. It means that $F([x_0, x_1]_{\varphi(t)})$ has value one until $t^*$ and zero afterwards, so the score $P$ defined in (2) is $P(\varphi) = t^*$. By the constant speed property, $t^* = \int_\Omega \varphi(t^*)$, so we end up maximising the mask area on the boundary.

# 5 Optimisation Problem and Algorithm

We proceed to define the optimisation problem at hand and how to solve it numerically.

Conceptually we try to find the ablation path (see Definition 3.1) that maximises the score $P(\varphi)$:

$$\max_{\varphi \in \mathcal{A}} P(\varphi).$$

Recall that the set $\mathcal{A}$ of ablation paths is convex; however, since the objective function $P$ is not convex, this is not a convex optimisation problem.

The method we suggest is to follow a gradient direction. Such an approach is in general not guaranteed to approximate a global maximum, but empirically it does manage that quite well here.

## 5.1 Gradient and Metric

Note that the differential is an element of $\mathcal{L}^1([0,1], R^*)$, so we need a map from that space to $\mathcal{L}^\infty([0,1], R)$. For now we assume that $\varphi \in \mathcal{L}^2([0,1], R)$ and $\mathrm{d}P \in \mathcal{L}^2([0,1], R^*)$.

However, we still need a covariance operator $K \colon R^* \to R$. In practice, we use a covariance operator associated to a smoothing operator. For a measure $\mu \in R^*$, $\langle K\mu, \theta \rangle \coloneqq \langle \mu, \int_\Omega k(\cdot - \mathbf{r})\theta(\mathbf{r})\,\mathrm{d}\mathbf{r} \rangle$, where $k$ is a suitable smoothing function, for instance $k(\mathbf{r}) = \exp(-\|\mathbf{r}\|^2/\sigma)$. This allows us to consider the *gradient* of $P$. Note that different choices of metric will influence on the algorithm.

Since the optimisation problem is *constrained* (since $\varphi$ is constrained by the requirements in Definition 3.1), following the gradient direction will lead us to violate the constraints. Since the constraints are convex, it is straightforward enough to project each gradient-updated version back to something that does fulfill them, and indeed that is the idea behind our algorithm, however in practice it does by itself not yield convergence without impractically many iterations. See Appendix C for the details of how we actually proceed.

# 6 Examples

To test out our path-scoring approach and the saliency method based on its optimisation, we use a standard image classifier deep-CNN (Inception v4 Szegedy et al. (2016), pre-trained on ImageNet), with a selection of real-world images for both the *current target* and *baseline* inputs. For each example pair, we compare multiple saliency methods.

Our algorithm yields a whole path of masks, which cannot as a whole be visualised in one 2D view. Generally, the *threshold mask*, which we define as the mask right where the path crosses the decision boundary, is most insightful: it still preserves enough of the image to be classified correctly, but no more. All of the images in this section refer to that selection, and where the threshold lies in the path is indicated by the vertical black line in the score plots. To be precise, this is the mask that preserves as little of the image as possible whilst still resulting in the intended class keeping the highest probability among the classes in the classifier's prediction. Note that although the threshold mask encapsulates a large part of our method's useful output, we find that the additional information from the rest of the path, and the score-plot along the path, also provide good diagnostics especially in examples where the mask highlights an unexpected region of the image.

Figure 2 is an example with particularly clear-cut interpretation: the vibrantly coloured rubber eraser is sufficient to retain the classification even in an almost completely ablated image. All of the compared methods manage to find a mask that focuses in on that, though with differences in the details. The unconstrained optimisation narrows it down to a few individual pixels, which gives an extremely good score (somewhat unsurprisingly, since that is what is being optimised), but the result is hardly useful for interpretability purposes: this should be considered an adversarial example. One interpretation of this is that identifying the gradient with the differential implies that the space of masks $R$ is essentially bounded functions wihout further regularity (see §5.1), similar to the mask space in §4.2. The region boundary in that space seems to be extremely close to the baseline, and the optimisation method finds those adversarial examples.

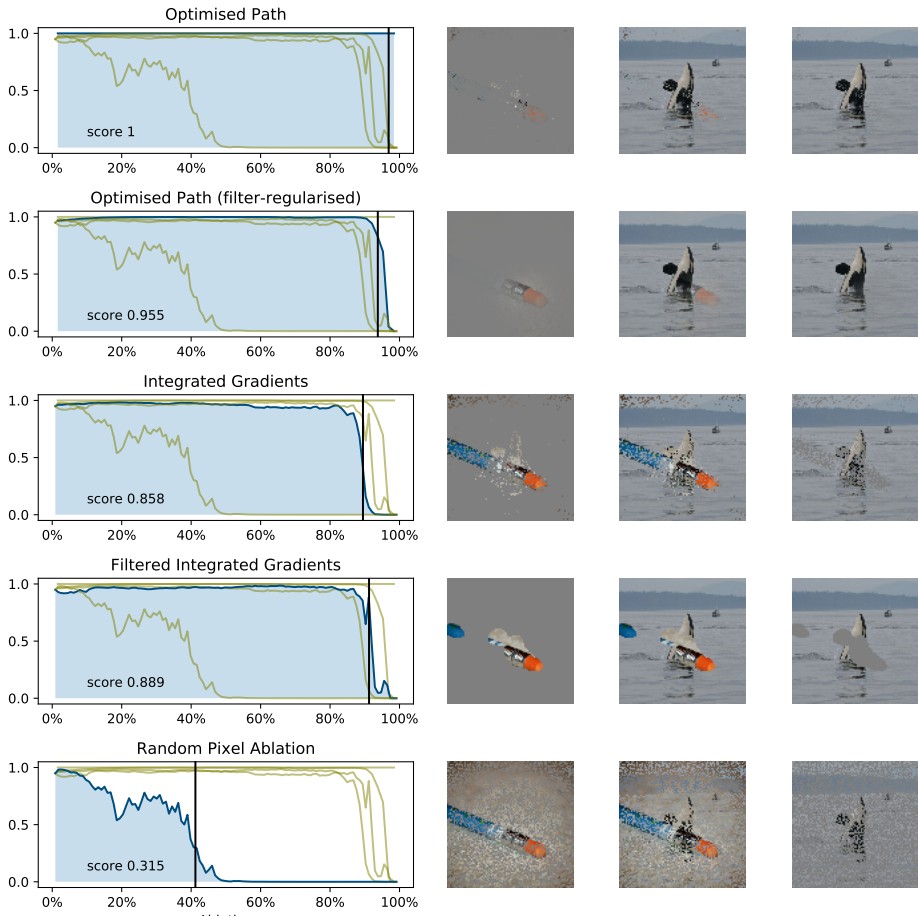

Figure 2: Comparison of saliency methods for an image of a pencil, against one of an orca whale as the baseline.

The integrated gradient method in principle also has this issue, and indeed the corresponding mask (pixel ranking, cf. §4.2) is quite grainy/noisy, without however behaving adversarially (the entire pencil is highlighted).

The authors in Fong & Vedaldi (2017) were confronted with similar problems. We use smoothing filters to compute the gradient, in order to both avoid adversarial examples and to have less noisy saliency masks. Using that, the path-optimisation still manages to achieve a high score, but now highlights the eraser as a single, clearly interpretable feature.

Filtering can also be applied to integrated gradient before pixel-ranking. That does avoid the noisyness, but it also leads to a *blobby* shape of the masks.

Comparison with a random-order ablation confirms that the good scores of the saliency methods really are a result of non-trivial feature highlighting, rather than something that comes out of the transition between the images automatically.[2] In Figure 3 it is evident that the saliency methods do *not* in general agree so clearly. Here, the filtered optimal path again highlights a small, single region that is sufficient for a classification of the image as a house- rather than goldfinch. Arguably, this is again adversarial (a human would certainly label the composite image gold- rather than house finch). However it does give rise to a useful interpretation: note that the highlighted region includes

---

[2]See Appendix E for a small, non-rigorous statistical study suggesting that scores $> 0.9$ are $p < 0.01$ significant against a null hypothesis of smoothly random masks, and extremely unlikely with pixelwise-random masks.

the house finch's plumage, whilst covering specifically the wing of the gold finch (which features a distinctive pattern not seen on house finches). So in this case, the saliency tells more about the baseline than about the current target.

The integrated gradient meanwhile hardly manages to mask out anything at all, before the classification switches to gold finch.

Practically speaking, saliency is particularly interesting in questions of whether a classification was actually correct. Figure 4 shows an example where the model misclassified an apple as a fig. The unstable scores in even the optimised ablation path are perhaps symptomatic of this (we are not sure how representative this is); nevertheless both our method and integrated gradients find a mask of less than half of the image that still retains the fig-classification. Whilst with integrated gradients, this mask includes the apple (which might in a real-world unclear situation lead one to think the classification payed attention to the intended target, increasing credibility), out method highlights mainly a patch of background at the border of the image.

The optimised paths depend not only on the model and current target, but also on the baseline; notice for example that the pencil, of which in Figure 2 mostly the eraser was highlighted, is in Figure 5 mostly characterised by the eraser's metallic joint as well as part of the blue shaft, which happens to coincide with the gold finch's wings that were also masked out in Figure 3. In Figure 6 it is something in between. Still the results (with filtering) tend to be relatively coherent across baselines, more so than with the Integrated Gradients or the adversarial unfiltered ones.

One might ask why to use a true image as a baseline at all (as opposed to a blurred image, a constant colour image, as in Sturmfels et al. (2020) or Fong & Vedaldi (2017)). The problem with artificial baselines is that the network missclassifies them (a blurred goldfinch is classified as a power drill, for example), so the ablation path crosses region where the network is extrapolating wildly. The resulting saliency may be difficult to interpret since the –unknown– parts of the baseline which the networks considers as important (which part of the blurred goldfinch lead to the power drill classification?) have an influence on the saliency of the current image. Ideally, we would like to have some result about saliency that gives good scores across many different baselines. Future research is needed.

Another choice to be made is the regularisation. We used here Gaußian filters; different sizes compared in Figure 6. It is a tradeoff between blurring out the boundaries and inviting noisiness, however even a small filter appears to be enough to avoid the algorithm from converging to adversarial examples (seemingly smaller than what Fong et al. (2019) require). It is even possible to scale down the filter during the optimisation to eventually give very sharp edges but not noise elsewhere, though it is somewhat dubious what sense this makes mathematically, from the point of view that the filtering represents just a metric on $R$. Again, further research is needed to assess the reliability.

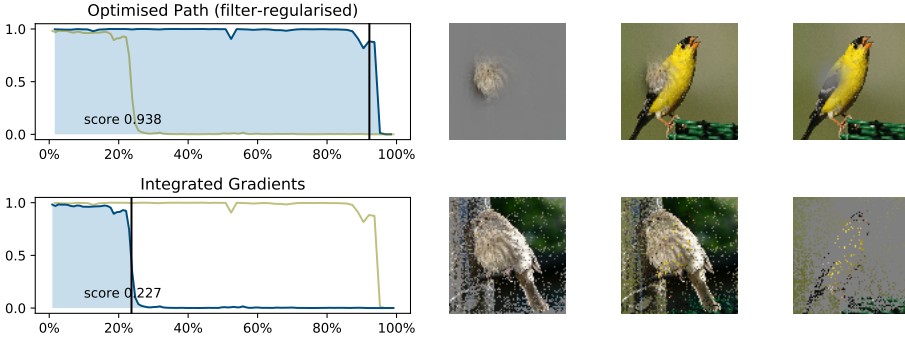

Figure 3: Method comparison for a house finch image against a gold finch baseline.

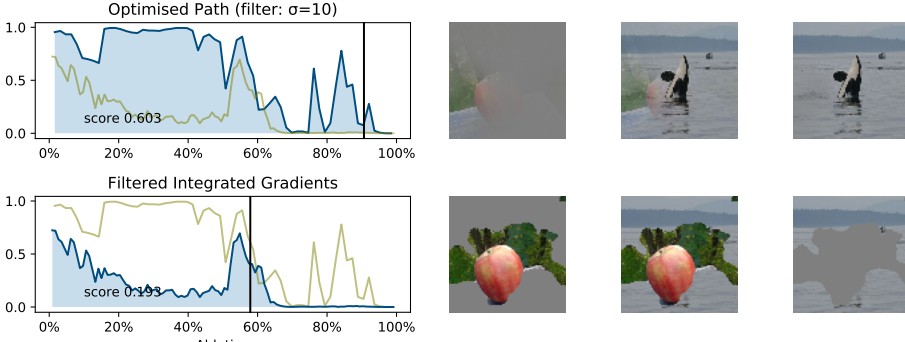

Figure 4: Method comparison for a misclassified image: Inception classifies the apple as a fig instead.

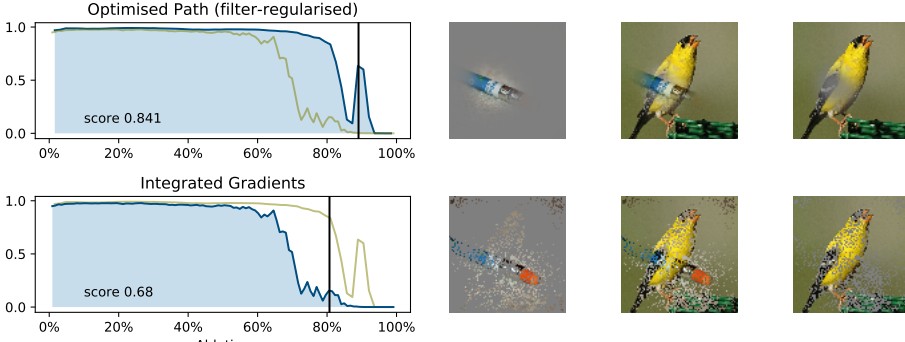

Figure 5: Comparison of saliency methods for an image of a pencil, against one of a gold finch as the baseline.

# 7 Pointing game

We evaluate our saliency algorithm using the pointing game method. This method was introduced in Zhang et al. (2017) and used, for instance, in Selvaraju et al. (2016); Fong & Vedaldi (2017). The primary goal is to show that our method, applied to a good enough image classifier, typically gives results that match what a human would also think of as the important part of the image. Specifically, we check whether the maximum-salient pixel lies within the bounding box of the object that defines the class.

Table 1 shows some results for our method on various images with blurred image as baseline. We show a few examples of this pointing game on Figure 7. See Appendix D for details and caveats with these results.

| Class (sample size) | Ablation path | Meaningful Perturbations |
|---|---|---|
| Bee (121) | 83 % | 46 % |
| Power drill (121) | 77 % | 40 % |
| Saxophone (119) | 89 % | 49 % |

Table 1: The success rate of the pointing game for various classes from the ILSVRC2014 dataset Russakovsky et al. (2015). We chose the first images in each class by alphabetical order. We compare to the meaningful perturbation method in Fong & Vedaldi (2017) (see caveat at the end of Appendix D).

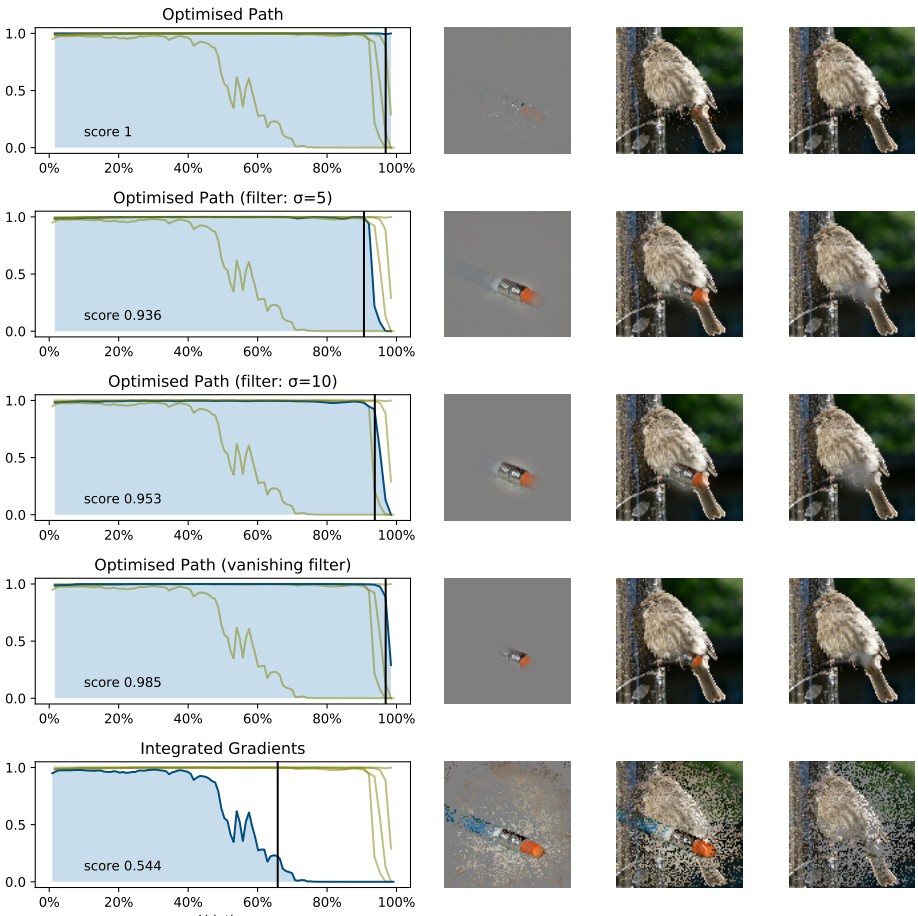

Figure 6: Comparison of saliency methods for an image of a pencil, against one of a house finch as the baseline.

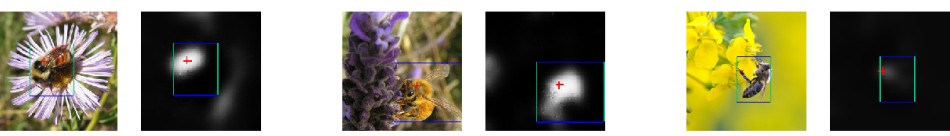

Figure 7: Examples of how the ablation path saliency "points" in the bounding box of various images of bees. The bounding boxes are human defined in the dataset, the red crosses indicate the location of least-ablated pixel.

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

## A    Canonical Time Reparametrisation

*Proof of Lemma 3.2.* The function $m \colon [0,1] \to \mathbb{R}$ defined by $m(t) \coloneqq \int_\Omega \varphi(t)$ is increasing and goes from zero to one (since we assume that $\int_\Omega 1 = 1$).

Note first that if $m(t_1) = m(t_2)$, then $\varphi(t_1) = \varphi(t_2)$ from the monotonicity property. Indeed, supposing for instance that $t_1 \leq t_2$, and defining the element $\theta \coloneqq \varphi(t_2) - \varphi(t_1)$ we see that on the one hand $\int_\Omega \theta = 0$, on the other hand, $\theta \geq 0$, so $\theta = 0$ and thus $\varphi(t_1) = \varphi(t_2)$.

Now, define $\mathsf{M} \coloneqq m([0,1]) = \{\, s \in [0,1] \mid \exists t \in [0,1] \, m(t) = s \,\}$. Pick $s \in [0,1]$.

If $s \in \mathsf{M}$ we define $\psi(s) \coloneqq \varphi(t)$ where $m(t) = s$ (and this does not depend on which $t$ fulfills $m(t) = s$ from what we said above). We remark that $\int_\Omega \psi(s) = \int_\Omega \varphi(t) = m(t) = s$.

Now suppose that $s \notin \mathsf{M}$. Define $s_1 \coloneqq \sup(\mathsf{M} \cap [0,s])$ and $s_2 \coloneqq \inf(\mathsf{M} \cap [s,1])$ (neither set are empty since $0 \in \mathsf{M}$ and $1 \in \mathsf{M}$). Since $s_1 \in \mathsf{M}$ and $s_2 \in \mathsf{M}$, there are $t_1 \in [0,1]$ and $t_2 \in [0,1]$ such

that $m(t_1) = s_1$ and $m(t_2) = s_2$. Finally define $\psi(s) := \varphi(t_1) + (s - s_1)\frac{\varphi(t_2) - \varphi(t_1)}{s_2 - s_1}$. In this case, $\int_\Omega \psi(s) = m(t_1) + (s - s_1)\frac{m(t_2) - m(t_1)}{s_2 - s_1} = s$. The path $\psi$ constructed this way is still monotone, and it has the constant speed property, so it is an ablation path. □

## B $\mathcal{L}^\infty$-optimal Monotonicity Projection

The algorithm proposed in Appendix C for optimising monotone paths uses updates that can locally introduce nonmonotonicity in the candidate $\hat{\varphi}_1$, so that it is needed to project back onto a monotone path $\varphi_1$. The following routine[3] performs such a projection in a way that is optimal in the sense of minimising the $\mathcal{L}^\infty$-distance[4], i.e.

$$\sup_t \left| \varphi_1(t, \mathbf{r}) - \hat{\varphi}_1(t, \mathbf{r}) \right| \leq \sup_t \left| \vartheta(t, \mathbf{r}) - \hat{\varphi}_1(t, \mathbf{r}) \right|$$

for all $\mathbf{r} \in \Omega$ and any other monotone path $\vartheta$.

The algorithm works separately for each $\mathbf{r}$, i.e. we express it as operating simply on continuous functions $p : [0, 1] \to \mathbb{R}$. The final step effectively *flattens out*, in a minimal way, any region in

---

**Algorithm 1** Make a function $[0, 1] \to \mathbb{R}$ nondecreasing

$\cup_i [l_i, r_i] \leftarrow \{\, t \in [0, 1] \mid p'(t) \leq 0 \,\}$      ▷ Union of intervals where $p$ decreases
**for** $i$ **do**
    $m_i \leftarrow \frac{p(l_i) + p(r_i)}{2}$
    $l_i \leftarrow \max\{\, t \in [r_{i-1}, l_i] \mid p(t) \leq m_i \,\}$
    $r_i \leftarrow \min\{\, t \in [r_i, l_{i+1}] \mid p(t) \geq m_i \,\}$
**end for**
**for** $i, j$ **do**
    **if** $[l_i, r_i] \cap [l_j, r_j] \neq \emptyset$ **then**
        **if** $m_j < m_i$, merge the intervals and recompute $m$ as the new center
    **end if**
**end for**
**return** $t \mapsto \begin{cases} p(t) & \text{if } t \notin \cup_i [l_i, r_i] \\ m_i & \text{if } t \in [l_i, r_i] \end{cases}$

---

which the function was decreasing.

In practice, this algorithm is executed not on continuous functions but on a PCM-discretised representation; this changes nothing about the algorithm except that instead as real numbers, $l, r$ and $t$ are represented by integral indices.

## C Path Optimisation Algorithm

As said in tsection § 5, our optimisation algorithm is essentially gradient descent of a path $\varphi$: it repeatedly seeks the direction within the space of all paths that (first ignoring the monotonicity constraint) would affect the largest increase to $P(\varphi)$ as per (2). As discussed before, this choice already requires a metric to obtain a vector-gradient from the covector-differential, which could be either the implicit $\ell^2$ metric on the discretised representation (pixels), or a more physical kernel/filter-based metric. We conceptually use the latter, however for technical reasons do not immediately apply the corresponding filter to the differential but rather to the *path*, which is not quite equivalent but does have the effect of avoiding noise from creeping into the state.

Unlike with the monotonisation condition, the update can easily be made to preserve speed-constness by construction, by projecting for each $t$ the gradient $\mathbf{g}$ on the sub-tangent-space of zero change to

---

[3] It is easy to come up with other algorithms for monotonising a (discretised) function. One could simply *sort the array*, but that is not optimal with respect to any of the usual function norms; or clip the derivatives to be nonnegative and then rescale the entire function, but that is not robust against noise pertubations.
[4] Note that the optimum is not necessarily unique.

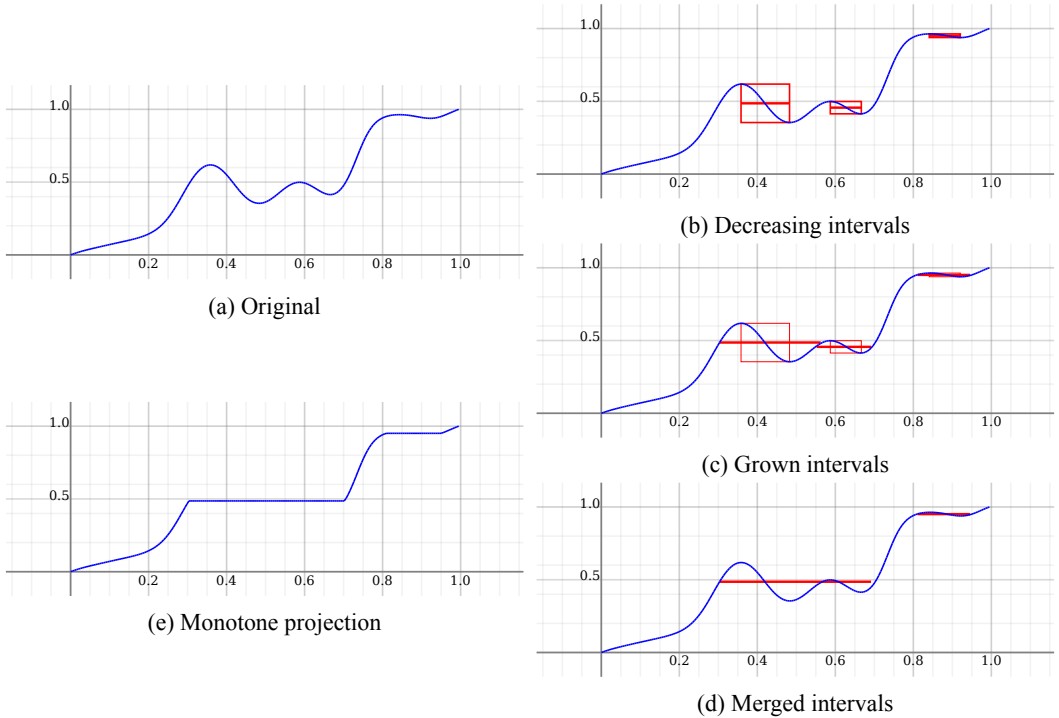

(a) Original

(b) Decreasing intervals

(c) Grown intervals

(d) Merged intervals

(e) Monotone projection

Figure 8: Example view of the monotonisation algorithm in practice. (a) contains decreasing intervals, which have been localised in (b). For each interval, the centerline is then extended to meet the original path non-decreasingly (c). In some cases, this will cause intervals overlapping; in this case merge them to a single interval and re-grow from the corresponding centerline (d). Finally, replace the path on the intervals with their centerline (e).

---

**Algorithm 2** Projected Gradient Descent

1: $\varphi \leftarrow ((t, \mathbf{r}) \mapsto t)$        ▷ Start with linear-interpolation path
2: **while** $\varphi$ is not sufficiently saturated **do**
3:      **for** $t$ in $[0, 1]$ **do**
4:          $x_{\varphi, t} := (1 - \varphi(t)) \, x_0 + \varphi(t) \, x_1$
5:          compute $F(x_{\varphi, t})$ with gradient $\mathbf{g} := \nabla F(x_{\varphi, t})$
6:          let $\hat{\mathbf{g}} := \mathbf{g} - \int_\Omega \mathbf{g}$        ▷ ensure $\hat{\mathbf{g}}$ does not affect mass of $\varphi(t)$
7:          update $\varphi(t, \mathbf{r}) \leftarrow \varphi(t, \mathbf{r}) - \gamma \, \langle \hat{\mathbf{g}}(\mathbf{r}) \mid |x_1 - x_0\rangle$ , for $\mathbf{r}$ in $\Omega$      ▷ $\gamma$ is learning rate
8:          (optional) apply a filter to $\varphi(t)$
9:      **end for**
10:      (optional) apply nonlinear gain to $\varphi$
11:      **for** $\mathbf{r}$ in $\Omega$ **do**
12:          re-monotonise $t \mapsto \varphi(t, \mathbf{r})$, using Algorithm 1
13:      **end for**
14:      clamp $\varphi(t, \mathbf{r})$ to $[0, 1]$ everywhere
15:      re-parametrise $\varphi$, such that $\int_\Omega \varphi(t) = t$ for all $t$ (using Appendix A)
16: **end while**.

---

$\int_\Omega \varphi(t)$, by subtracting the constant function times $\int_\Omega \mathbf{g}(t)$. Note this requires the measure of $\Omega$ to be normalised, or else considered at this point.

Then we apply these gradients time-wise as updates to the path, using a scalar product in the channel-space to obtain the best direction for $\varphi$ itself (as opposed to the corresponding image composit $x_{\varphi, t}$).

Simply projecting the updates path then again to the set of legal (in the sense of Definition 3.1) ablation paths would presumably enough to converge towards a saturated path with high score, however

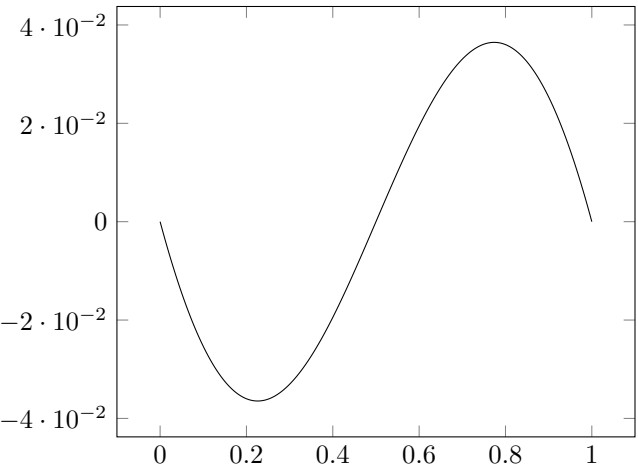

Figure 9: Function $\epsilon \colon [0,1] \to \mathbb{R}$. Note that $|\epsilon(x)| \le 5 \times 10^{-2}$. The formula is $\epsilon(x) :=$ $\tanh(2\zeta(x - 1/2))/2\tanh(\zeta) + 1/2 - x$, with $\zeta = 0.8$.

in tests with artificial constant gradient we found out that this requires extremely many iterations, with the number apparently scaling with the image dimension. The problem here is that saturating updates tend to be undone again by the speed-normalising reparametrisation, except for the most affected pixel. (Just increasing the learning rate does not help with this.) If the gradients come from a deep CNN and for every pixel the monotonicity needs to be restored, such many iterations would be prohibitly computation-intensive.

Fortunately we can dramatically speed up the convergence by artificially encouraging saturation. We tweak the ablation path $\varphi$ pointwise with a sigmoidal function that brings values lower than $\frac{1}{2}$ slightly closer to 0, and values greater than $\frac{1}{2}$ slightly closer to 1. To this end, we use a perturbation of the identity function defined by $x \mapsto x + \epsilon(x)$, and apply it pointwise to the path:

$$\varphi(\mathbf{r}, t) \leftarrow \varphi(\mathbf{r}, t) + \epsilon(\varphi(\mathbf{r}, t))$$

The perturbation function $\epsilon \colon [0,1] \to \mathbb{R}$ has the property that $x + \epsilon(x)$ maps $[0,1]$ into itself. The function $\epsilon$ we use is plotted on Figure 9. Although this perturbation seems small, the examples in §6 now only require 20-40 iterations.

As to the motivation behind the transformation $\varphi \to \varphi + \epsilon(\varphi)$, notice first that fully saturated masks – i.e., those that choose for every pixel are either zero or one, selecting exactly the value of either the current target or the baseline – are fixpoints of the function $x \mapsto x + \epsilon(x)$ since $\epsilon(0) = \epsilon(1) = 0$. So if such a mask is the optimum in the algorithm without artificial saturation (while there is no guarantee for this in general, this seems fairly frequent in practice), then it will also be an optimum of the algorithm with artificial saturation.

What is more, the dynamical system $x \mapsto x + \epsilon(x)$ quickly converges to zero or one, which efficiently encourages saturation, without sacrificing precision, as the function $\epsilon$ we chose is quite small.

Conversely, and unlike high learning rate, the saturation function is by construction monotone, symmetric and keeps the signal within the allowed bounds, so it avoids violating the ablation path constraints. The problem with high learning rates is that the algorithm gets caught in a sequence of alternating strong constraint-violating updates followed by a projection step that largely undoes the previous update.

## D   Evaluation with Pointing Game

The Pointing Game, the results of which we show in §7, is a way to verify that the saliency method points at a region that a human would consider relevant to the classification. It is often the region of an image that contains the physical object which is being classified.

Such evaluations of saliency methods clearly have caveats. One can for instance argue that the cases when the saliency points somewhere outside the bounding box are the most insightful ones, as they indicate that the classifier is using information from an unexpected part of the image (for instance, the background). Another caveat is that, if winning at the pointing game is the goal, a saliency method is only as good as its underlying classifier is. Nevertheless, if a saliency method often hits the bounding box it is reasonable to conclude that both the classifier and the saliency method behave in an intuitive way, from a human perspective. Our measurements in Table 1 confirm that the ablation-path method indeed does this well.

Our method has an advantage over other ones as it yields not only a single spatial map as the saliency but a whole path of masks. However, it does not directly give a saliency map. Here is how we choose one point in the image from an ablation path. First, we choose a specific time, just as we did in the examples in §6. In most cases, this means that we chose the time at which the probability has dropped by 20 %, that is the smallest time $t$ such that $F(t) \leq 0.8$. When there is no such time, we pick the one for which $F$ takes the maximum value on the path. Now that the time is chosen, we pick the point in the mask which have the smallest value. Note that, even though this selection may be unstable (since many points are close to the minimum value, as the mask is typically saturated), it usually does not matter since the whole region selected by the mask is salient.

The classifier in this experiment was an EfficientNet Tan & Le (2019) pre-trained on ImageNet, the test data set 360 images from three synsets out of ImageNet.

As a comparison, we evaluated the pointing game with the same network and inputs also through the closely related Meaningful Perturbations method Fong & Vedaldi (2017). We used a third-party PyTorch implementation Gildenblat (2017). Note that this implementation uses a slightly simplified method of removing information for performance, and that we did not change any of the hyperparameters of the method, so it is very likely that these results are not optimal. We also note that Fong & Vedaldi (2017) themselves include results for the pointing game on different data, with better scores.

Nevertheless, the point was to show that our method works reasonably well, and this may be due to the use of ablation paths: that increasing family of mask probably gives some stability to the saliency method, which improves the scores.

## E   Random ablation paths

In this section, following an idea of Sturmfels et al. (2020), we compare our method to random ablation-paths. Figure 2 includes one such example – the random path, which has a quite fuzzy class transition in roughly the middle of the path. Figure 10 shows that this typical behaviour. In

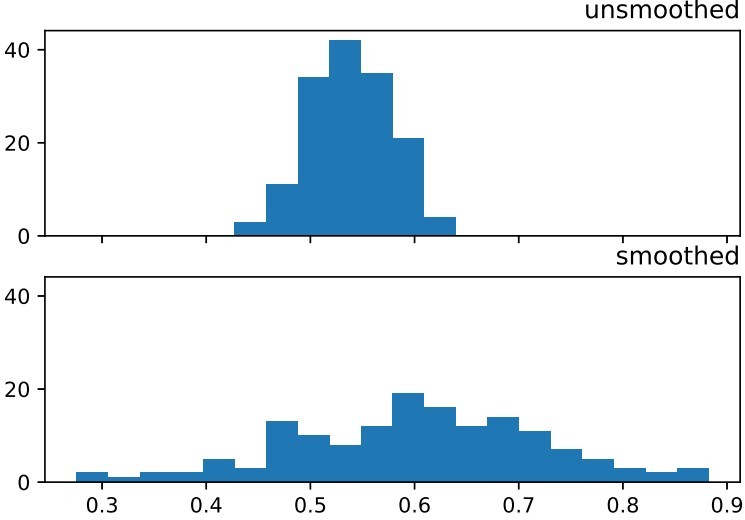

Figure 10: Histograms of the scores of ablating an image obtained as explained in §4.2 (the gold finch from Figure 3, against its blurred baseline) along random paths.

the unsmoothed case, the scores are almost compactly clustered in the middle, i.e., paths consisting of random single-pixel transitions rarely have exceptional scores. If the random paths are spatially smoothed, outliers become more likely (the paths could by coincidence mask out a whole particularly relevant region), but the scores are still really unlikely to reach 0.9, something our optimisation approach routinely attains.

## F  More examples

We provide in Figure 11 a series of ablation path results for one class in the ILSVRC14 datasetRussakovsky et al. (2015).

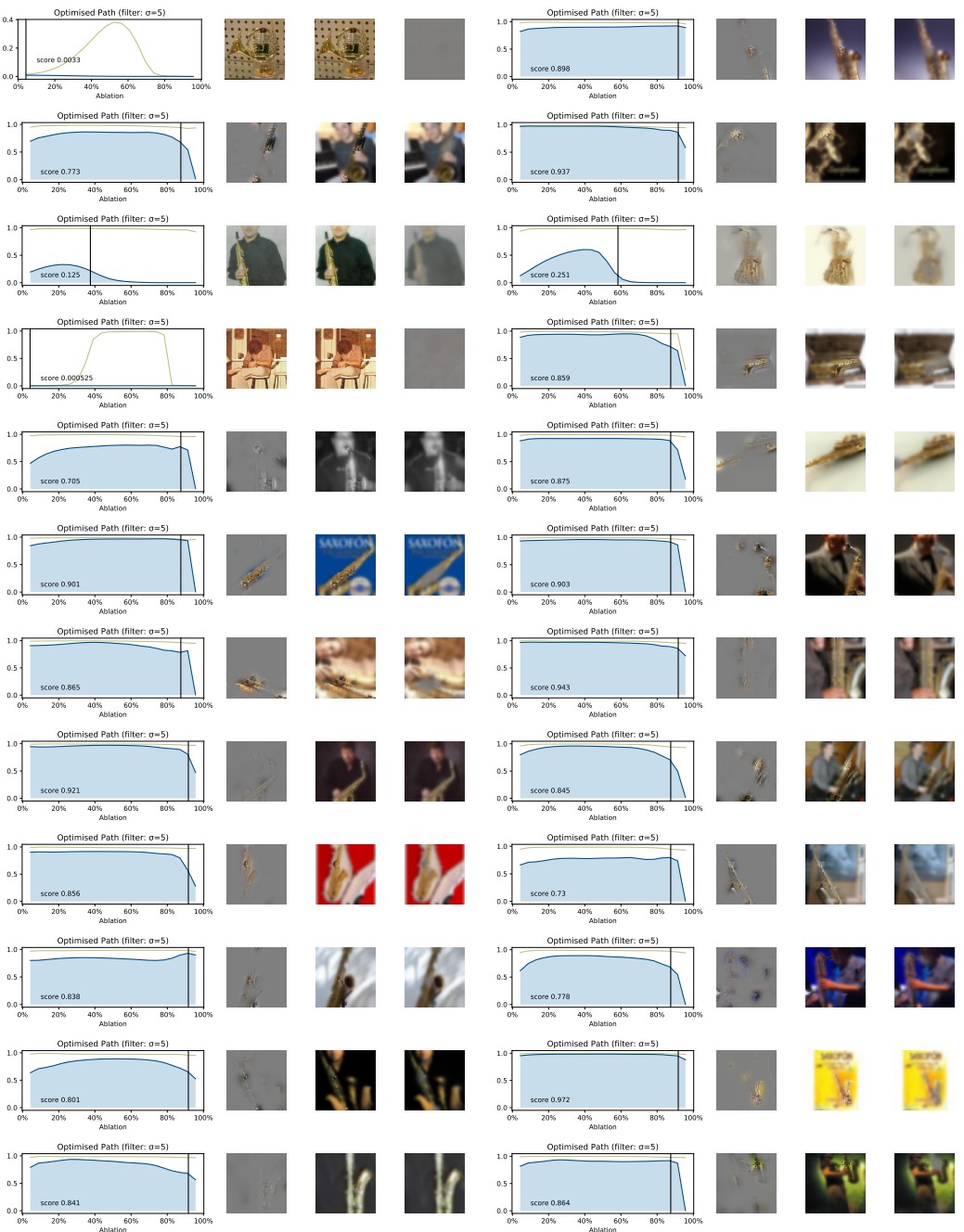

Figure 11: Some extra examples with EfficientNet and one class in the ILSVRC14 dataset. In some cases, the network does not correctly classify the image, which explains the poor score.

