# OpenReview forum: "Ablation Path Saliency"
_ICLR.cc/2021/Conference — Reject_

### Official Review · AnonReviewer1 · 2020-10-27
**Experimental evidence needed**

**Rating:** 4
**Confidence:** 4

**Review:**

The paper describes the black box approach to saliency prediction using ablation paths gradually replacing the parts of the image of one class with the image of another one. The idea has certain novelty, but the reviewer cannot see that it is backed by the evidence. The following comments are describing the reasons for the current paper rating and must be addressed for the rating to be improved:
1. The main concern of the reviewer about the paper is the lack of experimental evidence: it only provides the results for use cases and no quantitative assessment over the larger dataset.  For example, Fong&Vedaldi [1] assessed the method of 50K ImageNet validation images. As far as the reviewer can see, there is no qualitative assessment on a sufficient number of samples which would prove the efficiency and compare the method against the alternatives. This concern is critical for the given score of the paper.
2. Besides of that, it would be also important to emphasise the novelty: why do we need to have another saliency method, does it give any new insights other than the existing ones mentioned in the related work section?
3. It is also said in the paper: "We performed a small, non-rigorous statistical study suggesting that scores > 0.9 are p < 0.01 significant against a null hypothesis of smoothly random masks, and extremely unlikely with pixelwise-random masks.” It would be beneficial to see some further details of it as it could improve the analysis
4. In a light of the following description and the  given details of the optimisation method : “Since the optimisation problem is constrained (since φ is constrained by the requirements in Definition 3.1), following the gradient direction will lead us to violate the constraints. Since the constraints are convex, it is straightforward enough to project each gradient-updated version back to something that does fulfill them, and indeed that is the idea behind our algorithm, however in practice it does by itself not yield convergence without impractically many iterations” it is not clear, what the theoretical justification of it is. How do we know that the proposed artificial saturation could be generalised to other samples of data, beyond the use cases?
5. Further to that, the paper claims black box optimisation approach but the tests have only been reported for  Inception v4 pretrained on ImageNet. Would It still work if we use something different from it?
6. In a few qualitative examples given, the method is compared only to Integrated gradients. Is it possible to add a comparison with other methods,  e.g. Fong&Vedaldi[1] which is mentioned throughout the text as a closely related method?

[1] Fong&Vedaldi (2018) Interpretable Explanations of Black Boxes by Meaningful Perturbation

---

> ### Author Response · Authors · 2020-11-20
> **Reply to referee**
>
> > The main concern of the reviewer about the paper is the lack of experimental evidence:
>
> Yes, we agree with that criticism and we are working on testing our method on a large scale dataset for comparisons. The reason we did not do that in the first version is that our main focus was the formulation of an optimisation problem that unifies some existing saliency methods. We thus expect our method to be more amenable to interpretation and analysis, but not to give better scores in existing saliency metrics.
>
> > it would be also important to emphasise the novelty
>
> Our main point is not to provide a new saliency method, but rather to give a rigorous ground, and to unify, some of the existing saliency methods. By doing that, we indeed also propose a new saliency method. We argue that this method is easier to diagnose and interpret, due to its geometrical interpretation. The existing saliency method we cite in the related work section can be seen as approximate solutions to the optimisation problem we formulate.
>
> > It is also said in the paper: "We performed a small, non-rigorous statistical study [...]
>
> We will add some details on that in the appendix.
>
>
> > In a light of the following description and the given details of the optimisation method [...] How do we know that the proposed artificial saturation could be generalised to other samples of data, beyond the use cases?
>
> The artificial saturation will work in all other use cases where the input to the classifier is any tensor (images, sound, text with word embedding, etc.).
>
> You are right about the theoretical justifications of the artificial saturation algorithm, they are missing and we have now added some clarifications in the paper. In short: first, towards the end of the optimisation, the influence of the artificial saturation becomes negligible; second, in the beginning, it speeds up the training: the effect is similar to an increased learning rate, but without the undesirable effect dealing with projections of gradients which are too big.
>
>
> > Would It still work if we use something different from [Inception pretrained on ImageNet]?
>
> We can't see any reason why it should not work in other contexts, so our method should work regardless of the network or dataset. We will provide such examples in the revision.
>
> > Is it possible to add a comparison with other methods, e.g. Fong&Vedaldi[1] which is mentioned throughout the text as a closely related method?
>
> Yes, we are going to submit such a study in the revision.

---

### Official Review · AnonReviewer4 · 2020-10-28
**A saliency method that helps to understand the network decision**

**Rating:** 4
**Confidence:** 4

**Review:**

Summary:
Saliency problem for black-box classification is the main focus in this paper, which means to find out  the part of the image that is most relevant for the current model decision. Authors propose to find an optimal ablation path between two images to get such saliency maps. The finding in this paper suggest a new view based on ablation path optimization. Several examples are presented to show the behavior of the proposed method for one image classification model.

Pro:
+ Treating the saliency problem as finding an optimal ablation path between two images is interesting. Formulating the saliency problem as an optimization problem on paths is a kind of new.
+ Reasonable results can be obtained on several example images to show the decision of the model.

Concerns:
- The proposed ablation path saliency method requires the existence of example pairs, which limits the application of the method. It seems cannot handle the case if we are only interested in the saliency map for one single image without any support images (baseline image).
- It is also not clear how to select baseline images from a large image dataset (e.g. ImageNet) to get the optimal saliency map for one target image.
- There is a lack of quantitative evaluations to validate the effectiveness of the proposed method. Only showing several examples based on fixed image pairs is not that convincing. Why not follow some existing evaluation tools to conduct quantitative evaluations and compare with existing saliency methods? Evaluations could be Pointing Game or weakly-supervised localization/segmentation as in Grad-CAM (Selvaraju et al. 2016) as well as the sensitivity-n property proposed in (Ancona et al. 2017).
- Only examples for image classification models are presented. It is not clear whether the proposed method can be applied to other problems such as neural machine translation, question classification (Sundararajan et al. 2017), image captioning (Selvaraju et al. 2016) and etc.

---

> ### Author Response · Authors · 2020-11-20
> **Reply to referee**
>
> > The proposed ablation path saliency method requires the existence of example pairs, which limits the application of the method.
>
> Yes, this is in fact true of all the methods which require a baseline, in particular the method in Fong & Vedaldi, and the integrated gradient. First, our method actually works well with neutral baselines, for instance with blurred baselines. However, in the rare cases where the results are not as expected, we found it easier to diagnose the problem when the baseline is another image. The reason is that blurred images, or other such neutral baselines are in fact not neutral from the  network's point of view: these neutral images will be classified in some arbitrary class, with some corresponding, rather arbitrary, saliency. We are working on a multi-baseline method which would make the method baseline independent on average. This is very natural in our framework, as this is just a stochastic gradient applied to a slightly generalised optimisation problem.
>
>
>
>
> > It is also not clear how to select baseline images from a large image dataset (e.g. ImageNet) to get the optimal saliency map for one target image.
>
> Yes, we agree, we are working, as mentioned above, on a multiple-baseline version which we hope will address this question.
>
>
> > There is a lack of quantitative evaluations to validate the effectiveness of the proposed method.
>
> Yes, we agree, and we are now comparing our method to other existing ones using the pointing game approach.
>
> > [...] It is not clear whether the proposed method can be applied to other problems
>
> That is an excellent remark. We designed the framework with images in mind, but it works in all the examples that you mention, although we didn't test any of those. More precisely, our method works with any problem that can be formulated as a function from a vector space to a vector space, as that defines the module structure that we need.

---

> > ### Author Response · Authors · 2020-11-25
> > **Comment on the multiple-baseline idea**
> >
> > In the end, we decided against including results with several baseline for several reasons. The main reason is that using blurred baselines works quite well. On the other hand, our initial results for the multi-baseline approach are not yet convincing enough, so instead of pursuing this we directed our efforts towards addressing other issues raised by the referees.

---

### Official Review · AnonReviewer3 · 2020-10-29
**A good submission that aims at a valuable computer vision task.**

**Rating:** 6
**Confidence:** 3

**Review:**

The authors cast the saliency problem as finding an optimal ablation path between two images.
(1) According to the formulation of the saliency problem proposed in this study, the authors try to connect previous notions of saliency, notably integrated gradients and ablation tests via guiding where to add regularisation.
(2) One of my main concerns is that although the effect of the proposed strategy is illustrated based on several instances, more objective evaluation of specific tasks on large scale datasets needs to be compared to show if the formulated idea really works.
(3) As is known, the saliency mechanism is closely related to concepts in the field of neuroscience, so it would be better if the proposed formulation could be discussed based on its essences of cognitive neuroscience mechanisms.

---

> ### Author Response · Authors · 2020-11-20
> **Reply to referee**
>
>
> > [...] more objective evaluation of specific tasks on large scale datasets needs to be compared to show if the formulated idea really works.
>
> We agree that an objective evaluation is needed, and we are working in that direction. Nonetheless, our main emphasis was to connect existing, seemingly unrelated, saliency methods to a rigorous optimisation problem.
>
>
> > [...] it would be better if the proposed formulation could be discussed based on its essences of cognitive neuroscience mechanisms.
>
> This is a very interesting point of view indeed. It might very well be that cognitive mechanism follow an optimisation problem similar to the one we are proposing, but our expertise is too far from neuroscience for us to judge.

---

### Author Response · Authors · 2020-11-25
**Summary of the revision**

The changes in this revision are:
- A quantitative evaluation using the pointing game as metric on a few hundred images (we also compare to the meaningful perturbation approach by Fong & Vedaldi, although the implementation of their method that we use is probably suboptimal). (Section 7 and Appendix D)
- Clearer explanation and justification of the artificial saturation used to speed up the algorithm. It should now be clear that this modification should not significantly change the final result of the algorithm, as each step is a tiny modification of the gradient descent algorithm step. (Appendix C)
- We provide several examples with one other dataset and one other network architecture. More broadly, we expect our method to work with any of the standard network or datasets available. (Section 7)
- The extra tests are carried out with blurred versions of the image as baseline, which seems to work quite well. This shows that, as this choice is always available, it is not necessary to choose another image as baseline. (Section 7)
- We added details about the comparison of our method with a random saliency approach (Appendix E).
- We added a couple of raw examples (Appendix F)

---

### Decision · Program_Chairs · 2021-01-07
**Final Decision**

**Decision:**

Reject

**Comment:**

While the reviewers in general liked the ideas proposed in the paper, the experimental evaluation has several issues that need fixing before it can be accepted.